# The Significance of Systemic Immune-Inflammatory Index for Mortality Prediction in Diabetic Patients Treated with Off-Pump Coronary Artery Bypass Surgery

**DOI:** 10.3390/diagnostics12030634

**Published:** 2022-03-04

**Authors:** Tomasz Urbanowicz, Michał Michalak, Ahmed Al-Imam, Anna Olasińska-Wiśniewska, Michał Rodzki, Anna Witkowska, Assad Haneya, Piotr Buczkowski, Bartłomiej Perek, Marek Jemielity

**Affiliations:** 1Cardiac Surgery and Transplantalogy Department, Poznan University of Medical Sciences, 61-848 Poznan, Poland; anna.olasinska@poczta.onet.pl (A.O.-W.); michal.rodzki@skpp.edu.pl (M.R.); anna.witkowska2@skpp.edu.pl (A.W.); piotr.buczkowski@skpp.edu.pl (P.B.); bperek@ump.edu.pl (B.P.); mjemielity@poczta.onet.pl (M.J.); 2Department of Computer Science and Statistics, Poznan University of Medical Sciences, 61-806 Poznan, Poland; michal@ump.edu.pl (M.M.); ahmed.al.imam@student.ump.edu.pl (A.A.-I.); 3Department of Anatomy and Cellular Biology, College of Medicine, University of Baghdad, Baghdad 10047, Iraq; 4Herz and Gefaschirurgie, Universitatklinikum Schleswig-Holstein, 24105 Kiel, Germany; aassad.haneya@uksh.de

**Keywords:** type 2 diabetes mellitus, off-pump coronary artery bypass (OPCAB), systemic immune-inflammatory index (SII), survival analysis

## Abstract

Diabetes mellitus (DM) represents a complex carbohydrate metabolism disorder characterized by inflammatory over-reactivity. The study aimed to investigate the potential influence of postoperative inflammatory activation on mortality risk after off-pump coronary artery bypass grafting in diabetic patients. There were 510 patients treated with off-pump coronary artery bypass grafting due to stable complex coronary artery disease, including 175 patients with type-2 DM (T2DM.) The mean follow-up time was 3.7 +/− 1.5 years with a 9% all-cause mortality rate in the diabetic group. In multivariable analysis, preoperative comorbidities (stroke, peripheral artery disease, postoperative systemic inflammatory index >952, and postoperative left ventricle ejection fraction (LVEF) < 45%) were revealed as prognostic factors. The receiver operator characteristics curve analysis for postoperative calculations of systemic immune-inflammatory index (SII) appeared significant (AUC = 0.698, *p* = 0.008), yielding sensitivity of 68.75% and specificity of 71.07%. Systemic immune-inflammatory index (SII) can be regarded as a predictive marker for long-term prognosis in diabetic patients after off-pump coronary artery bypass grafting. The role of perioperative inflammatory activation may play a crucial role in mortality prediction.

## 1. Introduction

Diabetes mellitus (DM) is a complex carbohydrate metabolism disorder characterized by chronic hyperglycemia that eventually carries a high risk for cardiovascular morbidity [1]. The most common type 2 diabetes mellitus (T2DM) may induce atherosclerotic plaque development and take part in their progression [2]. 

Diabetes mellitus is claimed to be related to subclinical inflammatory activation as an underlying primary cause [3]. Inflammatory processes represent a chain of mechanisms triggering its manifestation; this complex metabolic disorder is linked with proinflammatory factor upregulation [4], including interleukin-1 and -6 elevated levels [5], tumor necrosis factor (TNF-alpha) [6], and alpha-1-acid glycoprotein [7]. In response, the endothelium damage markers are released locally or distributed with circulation inflammatory cells and are responsible for atherosclerosis development [8]. Vascular adhesion molecule (VCAM-1), intercellular adhesion molecule (ICAM-1), and e-selectins are involved in vascular permeability and inflammatory activation [9].

The trend revealing increased basal values of inflammatory markers and their overreaction to stimulus is already present in patients with disturbances of serum glycemic hemostasis, including diabetic state [10,11,12,13,14].

This glucose disorder is a significant risk factor for atherosclerosis development, including coronary artery disease [11]. Poor control of preoperative hyperglycemia in patients undergoing coronary artery bypass grafting may exert an adverse effect on long-term outcomes [12].

Diabetes mellitus is characterized by chronic hyperglycemic episodes that significantly impact physiologic coronary vasodilatation [13]. It contributes to microvascular endothelium-dependent coronary impairment through oxidative stress and activation of inflammatory cascades [13,14].

The increased risk for perioperative complications is postulated in association with inflammatory markers in surgical interventions [15,16].

Surgical revascularization is still the gold standard with the best outcomes for treating complex coronary disease [17]. The surgical intervention is related to activating inflammatory processes, mainly when performed with cardiopulmonary bypass [18]. Its avoidance through use of an off-pump technique is applied to minimalize systemic inflammatory response [19,20]. Moreover, the off-pump technique is believed to reduce perioperative neurological events, renal failure, blood product transfusions, length of the hospitalization period, and mortality [21,22].

The study aimed to investigate the potential influence of postoperative inflammatory activation on mortality risk after off-pump coronary artery bypass grafting in diabetic patients.

## 2. Materials and Methods

Between 2015 and 2018, 510 patients with complex coronary artery disease were operated on with off-pump coronary artery bypass grafting (OPCAB). The design of the study was single-center and retrospective. One hundred and seventy-five patients (42 (24%) females and 133 (76%) males) aged—years (median age: 67 (61–73) years) with concomitant type 2 diabetes mellitus (T2DM) and 335 patients (57 (17%) females and 278 (83%) males) aged—years (median age: 64 (59–70) years) without glucose metabolism impairment were operated on. The inclusion criteria for analysis are presented in Figure 1.

The diabetic and non-diabetic groups differed regarding age and comorbidities (arterial hypertension and hypercholesterolemia). Detailed clinical and demographical data are presented in Table 1.

The study was performed as a single-center retrospective analysis, approved by the Local Ethics Committee of Medical University of Poznan (approval number: 55/20 from 16 January 2020), and conducted following Good Clinical Practice principles and the Declaration of Helsinki.

The exclusion criteria for analysis included patients requiring concomitant valve surgery and those referred for surgery with inflammatory, autoimmune, oncological, or hematological proliferative diseases, acute coronary syndromes, and kidney failure.

The data of demographical, clinical, and laboratory parameters underwent statistical analysis. We used a routine hematology analyzer to assess the numbers of neutrophils, lymphocytes, monocytes, and platelets applied for systemic inflammatory response index (SIRI), systemic immune-inflammatory index (SII), and aggregate index of systemic inflammation (AISI) calculations (Sysmex Europe GmbH, Norderstedt, Germany).

The study end-point was all-cause mortality after the off-pump procedure. The analysis included death irrespective of the cause and was confirmed in the national database.

### Statistical Analysis

Data were presented as median and interquartile ranges Me (Q1–Q3), since data did not follow the normal distribution. A non-parametric test (Mann–Whitney) was used to compare interval variables for DM versus non-DM groups and DM deaths versus DM survivors. A receiver characteristic curve analysis was performed to find potential predictors for mortality. A proportional hazard regression model was used to find mortality risk factors. Both univariable and multivariable analyses (stepwise backward selection) were performed. The results were presented as hazard ratios (HR) and 95% confidence intervals (95% CI). The statistical analysis was performed using MedCalc statistical package; MedCalc^®^ Statistical Software version 20.010 (MedCalc Software Ltd., Ostend, Belgium; https://www.medcalc.org). All tests were considered significant at *p* < 0.05.

## 3. Results

Our study sample comprises 510 consecutive patients, including 175 (34%) with concomitant T2DM who are strictly on insulin, who were treated with off-pump coronary artery bypass grafting between January 2015 and December 2018 in our hospital. The indication for surgery included left main disease (178 pts (35%)) and two and three vessels disease (163 pts (32%) and 168 pts (33%)), respectively. The T2DM group was characterized by Hb1Ac preoperative results of 6.7 (6.5–6.9)% and required an average 21 +/− 4 IU total insulin daily, including 27 +/− 5% basal dose.

The mean surgery (skin-to-skin) time and mean number of performed anastomoses were 2.3 ± 0.4 h and 2.3 ± 0.2, respectively. The EuroScore II was calculated exclusively in in each patient and presented no significant differences between groups: 1.5 +/− 0.7 vs. 1.6 +/− 0.8 in non-DM and DM groups (*p* = 0.672), respectively.

The 30-days mortality was 1%. During the median 3.7 +/− 1.5 years follow-up, there was a 91% cumulative survival rate.

Patients were discharged on pharmacological treatment including statins, antiplatelets, B-blockers, and angiotensin-converting enzyme inhibitors, whenever tolerable.

### 3.1. All Groups

There was no difference between the diabetic and non-diabetic groups regarding preoperative and postoperative characteristics except for the postoperative serum creatinine concentration (*p* = 0.048) (Table 2).

### 3.2. Diabetic Group

Patients with T2DM were divided into two subgroups: survivors (*n* = 159) and deceased (*n* = 16). We presented a statistically significant difference in preoperative comorbidities, including chronic pulmonary obstructive disease (COPD) (*p* = 0.014), stroke (*p* < 0.001), and peripheral artery disease (PAD) (*p* = 0.010). Postoperatively, only neutrophil count (*p* = 0.007), SII (*p* = 0.008), and AISI (*p* = 0.045) were revealed as significant. There was no significant difference in maximum values of troponin-I (*p* = 0.539). Detailed characteristics are presented in Table 3.

### 3.3. Receiver Operator Characteristics for Postoperative Inflammatory Markers Revealed in Multivariable Analysis

We compared systemic immune-inflammatory indexes (SII) obtained from preoperative and postoperative data. The receiver operator characteristics curve (ROC) was performed for preoperative values as presented in Figure 2 (AUC = 0.643, *p* = 0.035), yielding sensitivity of 75% and specificity of 54.72% with a cut-off value of 665.

The ROC analysis for postoperative calculations of SII is presented in Figure 3 (AUC = 0.698, *p* = 0.008), yielding sensitivity of 68.75% and specificity of 71.07% with a cut-off value above 952.

### 3.4. Receiver Operator Curve for Postoperative Inflammatory Markers, including Components of the SII

The systemic immune-inflammatory index (SII) is expressed as a quotient of neutrophils and platelets divided over lymphocyte counts. We present, in Figure 4, components of the mentioned index comparing its prediction properties, although they were not found to be significant in multivariable analysis.

As presented in Figure 4, all components of the systemic immune-inflammatory index (SII) presented as single parameters possess lower predictive values: neutrophil count (AUC = 0.701, *p* < 0.001; yielding sensitivity of 93.75% and specificity of 42.14% with a cut-off value above 4.6), platelets (AUC = 0.514, *p* = 0.853; yielding sensitivity of 75% and specificity of 36.38% with a cut-off value below 303), and lymphocytes (AUC = 0.646, *p* = 0.036; yielding sensitivity of 43.75% and specificity of 82.39% with a cut-off value below 1.4), respectively.

### 3.5. Univariable Analysis

The univariate Cox regression analysis detected a significant effect of age (HR = 1.09, 95% CI 1.00–1.17, *p* = 0.027), and among comorbidities, including stroke (HR = 7.25, 95% CI 2.47–21.27, *p* < 0.001) and peripheral artery disease (PAD) (HR = 4.35, 95% CI 1.51–12.55, *p* = 0.050); these are significant factors for the long-term survival in T2DM patients. The postoperative laboratory parameters which were pointed out by the univariable analysis were: SII as binary variable indicating values above 952 (HR = 5.23, 95% CI 1.64–16.68, *p* = 0.005) and SIRI both as continuous and binary variable indicating values above 1030 (HR = 4.00, 95% CI 1.39–11.55, *p* = 0.010) as presented in Table 4.

### 3.6. Multivariable Analysis

The multivariable model included parameters revealed as significant in the DM group in univariable analysis. The multivariable analysis confirmed the significance of the following comorbidities: stroke (HR = 3.39, 95% CI 1.06–10.91, *p* = 0.040) and peripheral artery disease (HR = 3.83, 95% CI 1.27–11.56, *p* = 0.017). The systemic immune-inflammatory index SII > 952 (HR = 3.44, 95% CI 1.02–11.66, *p* = 0.047) was found to be the only postoperative laboratory result impacting the long-term prognosis after off-pump surgery. Postoperative left ventricle ejection fraction was the only significant echocardiographic parameter (HR = 4.11, 95% CI 1.21–13.95, *p* = 0.023) as presented in Table 5.

## 4. Discussion

The main finding of our study is highlighting the inflammatory reaction activation represented by the systemic immune-inflammatory index (SII) and its significance for long-term survival after off-pump coronary artery bypass grafting in diabetic patients treated due to complex coronary artery disease. The multivariable analysis revealed a predictive effect of clinical preoperative factors referred to in advanced atherosclerosis that resulted in a history of stroke episodes and peripheral artery disease. Moreover, a postoperative left ventricle ejection fraction below 45% was an indicator of mortality. Different inflammatory parameters obtained from whole blood count were taken into consideration, presenting exclusively the systemic immune-inflammatory index (SII) as significant in T2DM patients, which is novel regarding previous reports [23,24]. Despite common assumptions in clinical practice, the significance of inflammatory reactions rather than the myocardial perioperative injury markers was confirmed to be prognostic for long-term survival in the analysis [25,26].

The presented analysis is based on results obtained from elective procedures performed with the utility of the off-pump technique. We aimed to select a homogenous group of patients for better interpretation of data. Thus, the surgical interventions presented in acute coronary syndromes were excluded from analysis as a significant increase of inflammatory markers in urgent situations [27] was already postulated [28]. Another exclusion factor included surgical revascularization performed with cardiopulmonary bypass administration. The surgery performed on-pump as a standard technique is burdened with a high risk for systemic inflammatory response. The cardiopulmonary bypass was claimed to activate a defensive response to its components and non-physiological blood flow created by roller pumps [27,28]. The concomitant valve pathology requiring surgical interventions was excluded since combined procedures carry increased mortality risk [29,30].

To our best knowledge, our study is the first to present the systemic immune-inflammatory index (SII) as a mortality prediction marker for complex coronary artery disease treated by surgical revascularization in the off-pump technique.

As a lipid-driven inflammatory disease, atherosclerosis emerges from an imbalance between proinflammatory and inflammation-resolving mechanisms. Different markers can represent the inflammatory response, including the systemic immune-inflammatory index, which has gained interest in recent studies as a reliable predictor [31]. We found preoperative SII values above 665 to be significant in the univariable model for mortality prediction. Our results agree with Liu et al. (2021), who estimated the SII value of 653 to predict the severity of coronary stenosis [32]. According to the study, as mentioned earlier, the SII-predicting properties were higher than other parameters, including c-reactive protein (CRP) and neutrophil to lymphocyte and platelets to lymphocyte ratios (NLR, PLR). The SII reduction by dietary changes before surgical revascularization and its beneficial effect on adverse cardiovascular incidents were presented in Szymanska et al. (2021) [33].

There is no available data regarding standard values of inflammatory markers such as SII. The results obtained from the Rotterdam Study suggested 455 as a reference level [34] and were far below the values obtained from our patients that can be explained by complex coronary disease. Interestingly, the SII postoperative results were significantly higher in diabetic patients with poor long-term outcomes. The presented results indicate that the subgroup of patients with concomitant diabetes mellitus type 2, with postoperative SII results above 952, undergoing coronary artery bypass grafting, should be more scrupulously monitored in the long-term follow-up. The inflammatory reactions in the early reperfusion period, represented by a systemic inflammatory index, significantly influence all-cause mortality in off-pump patients. The commonly applied mortality risk scores, such as EuroScore II or STS score, reflect either perioperative mortality and/or morbidity [35,36]; these, however, are mainly valid for the early postoperative observation. The inflammatory activation seems to be the clue for identifying those patients undergoing the off-pump procedure outwardly uneventfully, who are, in fact, at greater risk for long-term mortality.

Our retrospective study highlights the effect of inflammatory reaction measured by the systemic immune-inflammatory index (SII) during the immediate postoperative period on long-term mortality prediction, which may explain its triggering characteristics for further atherosclerotic progression. The inflammatory cells and phenomena involved in atheroscleromatic disease burden include neutrophils, monocytes, and macrophages activation. Currently, there is much interest in the role of endothelial-cell-derived extracellular vesicles in the progression of atherosclerosis. Assuredly, the inflammatory status, as an initiator and active participant of atherosclerosis development, should be taken into consideration as an important issue for monitoring and a potential therapeutic target [37,38].

The inflammatory system overreaction after off-pump revascularization has been proven to influence the mortality rate [39]. The combination of preoperative clinical factors and postoperative inflammatory parameters contrary to postoperative myocardial injury markers possesses predictive values for mortality [40,41].

The study limitations refer its retrospective design and a single-center experience. This was a retrospective all-cause mortality analysis based on perioperative inflammatory reaction characteristics on the first postoperative day as an early reperfusion phenomenon. Moreover, we did not obtain data on cause of death, which would provide valuable additional information. Moreover, knowledge on diabetes management and coronary artery disease pharmacological treatment in the individual patient would be beneficial; however, this is challenging in the long-term treatment of large study groups.

## 5. Conclusions

Postoperative systemic immune-inflammatory index (SII) may be regarded as a potential index for long-term prognosis in diabetic patients treated by off-pump coronary artery bypass grafting. Perioperative inflammatory activation may play a crucial role in mortality prediction.

## Figures and Tables

**Figure 1 diagnostics-12-00634-f001:**
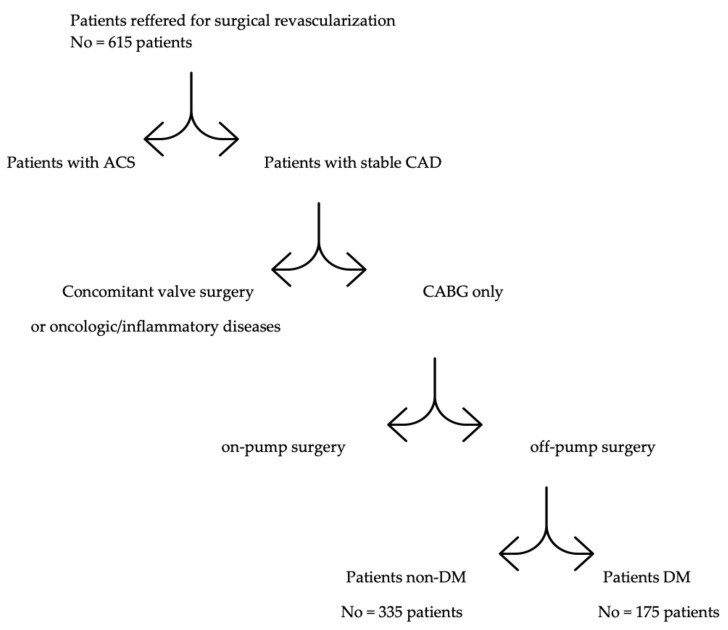
Inclusion criteria for retrospective analysis.

**Figure 2 diagnostics-12-00634-f002:**
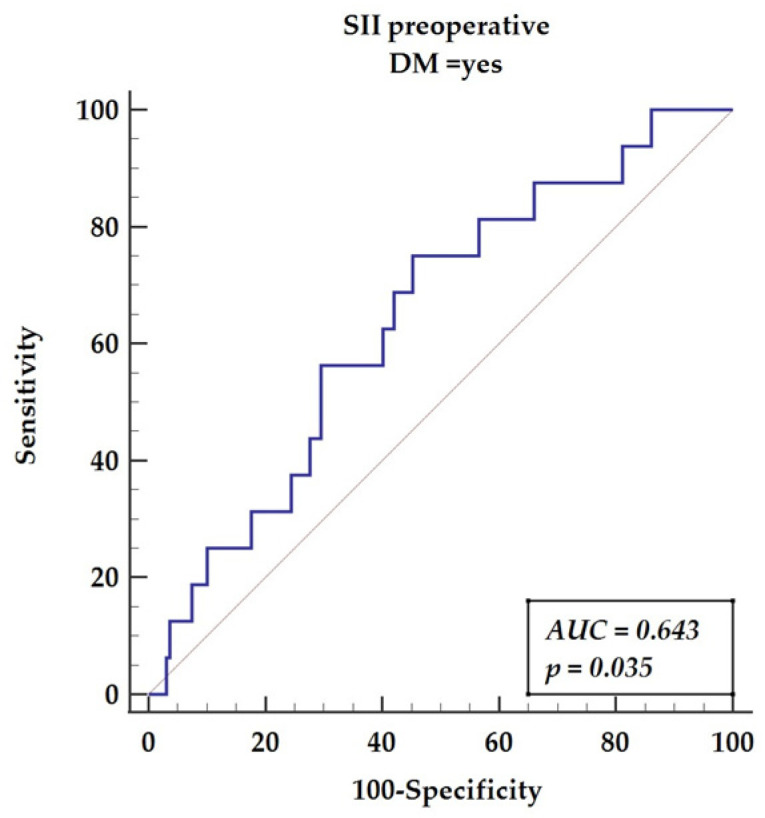
Receiver operating characteristics curve for preoperative SII in DM patients. Abbreviations: AUC—area under the curve, DM—diabetes mellitus, SII—systemic immune-inflammatory index.

**Figure 3 diagnostics-12-00634-f003:**
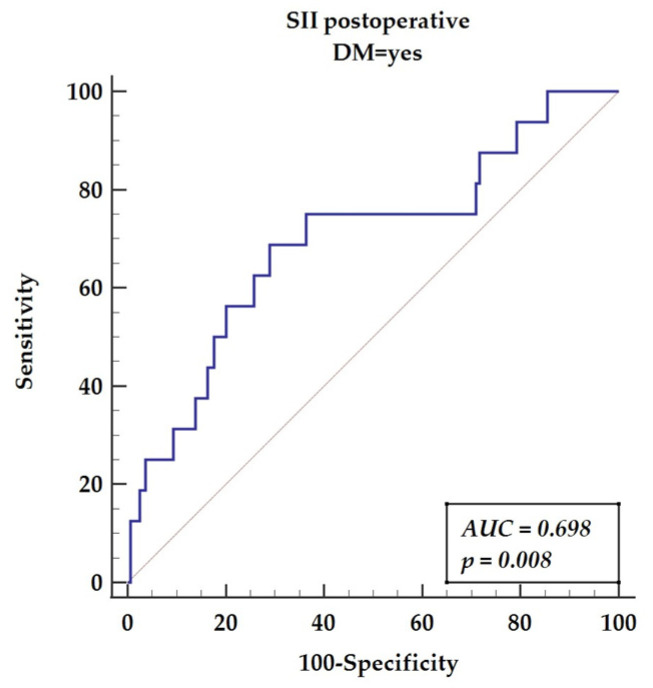
Receiver operating characteristics curve for postoperative SII in DM patients. Abbreviations: AUC—area under the curve, DM—diabetes mellitus, SII—systemic immune-inflammatory index.

**Figure 4 diagnostics-12-00634-f004:**
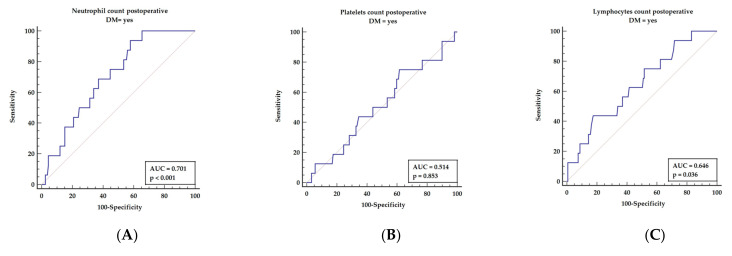
Postoperative simple components of SII index as predictive factors for long-term survival in Cox analysis. Abbreviations: AUC—area under the curve, (**A**) neutrophils, (**B**) platelets, (**C**) lymphocytes. Lymp—lymphocytes count, Neu—neutrophil count, Plt—platelets count.

**Table 1 diagnostics-12-00634-t001:** Demographical and clinical preoperative characteristics.

Parameters	DM Group (No. = 175)	Non-DM Group (No. = 335)	*p*
Demographical data:			
1. Age (years) (median (Q1–Q3))	67 (61–73)	64 (59–70)	0.002 *
2. Gender (F (%)/M (%))	42 (24%)/133 (76%)	57 (17%)/278 (83%)	0.058
Comorbidities:			
1. Arterial hypertension (median (Q1–Q3))	36 (21%)	97 (29%)	0.041 *
2. COPD (median (Q1–Q3))	12 (97%)	32 (10%)	0.716
3. Hypercholesterolemia (median (Q1–Q3))	77 (44%)	219 (65%)	<0.001 *
4. PAD (median (Q1–Q3))	27 (15%)	52 (16%)	0.967
Echocardiographic estimation of LVEF:			
1. Preoperative (%) (median (Q1–Q3))	55 (50–60)	55 (50–60)	0.349
2. Postoperative (%) (median (Q1–Q3))	55 (50–60)	60 (50–60)	0.773

Abbreviations: COPD—chronic obstructive pulmonary disease, DM—diabetes mellitus, F—females, LVEF—left ventricle ejection fraction, M—males, PAD—peripheral artery disease; *—statistically significant.

**Table 2 diagnostics-12-00634-t002:** Laboratory results: comparison between both groups (DM vs. non-DM).

Parameters	DM Group (No. = 175)	Non-DM Group (No. = 335)	*p*
Preoperative laboratory results:			
A. Whole blood count:			
1. WBC × 10^9^/L (median (Q1–Q3))	7.7 (6.5–9.3)	7.8 (6.4–9.2)	0.979
2. Lymphocyte × 10^9^/L (median (Q1–Q3))	1.8 (1.5–2.3)	1.8 (1.4–2.2)	0.415
3. Neutrophils × 10^9^/L (median (Q1–Q3))	5.0 (4.0–6.2)	5.1 (4.0–6.3)	0.748
4. Monocytes × 10^9^/L (median (Q1–Q3))	0.5 (0.4–0.6)	0.5 (0.4–0.6)	0.828
5. Hemoglobin × 10^9^/L (median (Q1–Q3))	8.6 (8–9.2)	8.7 (8.3–9.3)	0.147
6. Platelets × 10^3^/μL (median (Q1–Q3))	225 (189–270)	228 (192–264)	0.819
B. Hematologic indexes:			
1. SIRI (median (Q1–Q3))	1.3 (0.8–1.9)	1.3 (0.9–1.9)	0.483
2. SII (median (Q1–Q3))	634 (431–916)	619 (425–914)	0.649
3. AISI (median (Q1–Q3))	280 (178–470)	287 (172–440)	0.764
C. Myocardial injury:			
Troponin on admission × 10^9^/L (median (Q1–Q3))	0.01 (0.01–0.03)	0.01 (0.01–0.02)	0.527
D. Kidney function:			
Creatinine × 10^9^/L (median (Q1–Q3))	86 (76–106)	85 (71–99)	0.096
Postoperative laboratory results (24 h):			
A. Whole blood count:			
7. WBC × 10^9^/L (median (Q1–Q3))	8.6 (6.9–10.3)	8.4 (7–10.4)	0.925
8. Lymphocyte × 10^9^/L (median (Q1–Q3))	1.9 (1.5–2.5)	1.9 (1.5–2.5)	0.983
9. Neutrophils × 10^9^/L (median (Q1–Q3))	5.1 (3.8–6.6)	4.9 (3.7–6.5)	0.613
10. Monocytes × 10^9^/L (median (Q1–Q3))	0.84 (0.7–1.1)	0.9 (0.7–1.1)	0.587
11. Hemoglobin mmol/L (median (Q1–Q3))	6.9 (6.6–7.3)	6.8 (6.5–7.2)	0.082
12. Platelets × 10^3^/μL (median (Q1–Q3))	273 (227–341)	283 (229–350)	0.662
B. Hematologic indexes:			
4. SIRI (median (Q1–Q3))	4.3 (2.6–7.1)	4.2 (2.8–6.0)	0.556
5. SII (median (Q1–Q3))	699 (507–1062)	741 (497–1096)	0.846
6. AISI (median (Q1–Q3))	616 (391–1030)	634 (390–1088)	0.729
C. Myocardial injury:			
Troponin on admission μmg/L (median (Q1–Q3))	1.5 (0.7–3.5)	1.6 (0.7–3.8)	0.615
D. Kidney function:			
Creatinine mmol/L (median (Q1–Q3))	96 (79–124)	86 (71–107)	0.048 *
Number of performed grafts (median (Q1–Q3))	2.3 (2.0–2.6)	2.3 (2.1–2.5)	0.782
All-cause mortality during observation time (%)	16 (9%)	27 (8%)	0.817

Abbreviations: AISI—aggregate index of systemic inflammation, DM—diabetes mellitus, SII—systemic immune-inflammatory index, SIRI—systemic inflammatory response index, WBC—white blood count; *—statistically significant.

**Table 3 diagnostics-12-00634-t003:** Deaths and survivor group characteristics among T2DM patients.

Parameters	DM Deaths (No. = 16)	DM Survivors (No. = 159)	*p*
Age (years) (median (Q1–Q3))	70 (65–76)	67 (61–73)	0.097
Gender (F (%)/M (%))	5 (31%)/11 (69%)	37 (23%)/122 (77%)	0.476
Comorbidities:			
1. Arterial hypertension (%)	2 (13%)	34 (21%)	0.402
2. COPD (%)	4 (25%)	11 (7%)	0.014 *
3. Stroke (%)	6 (38%)	7 (4%)	<0.001 *
4. Hypercholesterolemia (%)	6 (38%)	71 (45%)	0.583
5. PAD (%)	6 (38%)	21 (13%)	0.010 *
Preoperative:			
1. WBC × 10^9^/L (median (Q1–Q3))	7.6 (6.9–8.8)	7.73 (6.5–9.3)	0.709
2. Lymphocytes × 10^9^/L (median (Q1–Q3))	1.6 (1.1–2.0)	1.8 (1.5–2.3)	0.105
3. Neutrophils × 10^9^/L (median (Q1–Q3))	5.5 (4.4–6.5)	5 (4.0–6.2)	0.399
4. Monocytes × 10^9^/L (median (Q1–Q3))	0.5 (0.3–0.5)	0.5 (0.4–0.6)	0.589
5. Hemoglobin mmol/L (median (Q1–Q3))	8.4 (7.5–9.4)	8.6 (8.1–9.2)	0.409
6. Platelets × 10^3^/μL (median (Q1–Q3))	234 (209–278)	224 (189–270)	0.566
Postoperative (24 h):			
1. WBC × 10^9^/L (median (Q1–Q3))	9.3 (7.6–12)	8.4 (6.7–10)	0.107
2. Lymphocytes × 10^9^/L (median (Q1–Q3))	1.7 (1.2–2.1)	1.9 (1.6–2.5)	0.055
3. Neutrophils × 10^9^/L (median (Q1–Q3))	6.2 (5–7.9)	5 (3.7–6.4)	0.007 *
4. Monocytes × 10^9^/L (median (Q1–Q3))	1 (0.6–1.1)	0.8 (0.7–1.1)	0.798
5. Hemoglobin mmol/L (median (Q1–Q3))	6.9 (6.6–7.2)	6.9 (6.6–7.3)	0.881
6. Platelets × 10^3^/L (median (Q1–Q3))	271 (231–327)	273 (227–341)	0.854
Preoperative indexes:			
1. SIRI (median (Q1–Q3))	1.5 (1–2.2)	1.2 (0.8–1.9)	0.223
2. SII (median (Q1–Q3))	839 (611–1068)	617 (422–904)	0.059
3. AISI (median (Q1–Q3))	355 (217–537)	267 (172–450)	0.183
Postoperative indexes:			
1. SIRI (median (Q1–Q3))	5.5 (3.5–9.6)	4.1 (2.6–6.7)	0.150
2. SII (median (Q1–Q3))	1097 (679–1956)	686 (495–1015)	0.008 *
3. AISI (median (Q1–Q3))	1094 (495–1704)	602 (358–978)	0.045 *
Troponin-I:			
1. On admission mcg/L (median (Q1–Q3))	0.01 (0.01–0.01)	0.01 (0.01–0.03)	0.867
2. Maximum mcg/L (median (Q1–Q3))	2.6 (0.6–5.0)	1.5 (0.7–3.1)	0.539
Creatinine			
1. Preoperative mmol/L (median (Q1–Q3))	97 (83–136)	86 (75–102)	0.232
2. Postoperative mmol/L (median (Q1–Q3))	102 (84–145)	96 (78–116)	0.351
LVEF:			
1. Preoperative (%) (median (Q1–Q3))	55 (50–60)	47 (43–51)	0.504
2. Postoperative (%) (median (Q1–Q3))	55 (45–55)	60 (50–60)	0.091

Abbreviations: AISI—aggregate index of systemic inflammation, COPD—chronic pulmonary obstructive disease, DM—diabetes mellitus, LVEF—left ventricle ejection fraction, PAD—peripheral artery disease, SII—systemic immune-inflammatory index, SIRI—systemic inflammatory response index, WBC—white blood count; *—statistically significant.

**Table 4 diagnostics-12-00634-t004:** Cox regression univariable analysis for long-term survival.

Parameter	HR	95% CI	*p*-Value
Demographical and clinical:			
1. Age	1.09	1.01–1.17	0.027 *
2. Stroke	7.25	2.47–21.27	<0.001 *
3. PAD	4.35	1.51–12.55	0.050 *
Preoperative laboratory parameters:			
1. SIRI	1.03	0.79–1.34	0.830
2. SII	1.00	1.00–1.00	0.555
3. SII > 665	2.45	0.77–7.85	0.131
4. AISI	1.00	0.99–1.01	0.896
Preoperative laboratory parameters:			
1. Neutrophils postoperatively	1.24	1.05–1.46	0.010 *
2. SIRI	1.07	0.94–1.20	0.299
3. SII	3.43	1.00–1.00	0.001 *
4. SII > 952	5.23	1.64–16.68	0.005 *
5. AISI	1.00	0.99–1.00	0.091
6. AISI > 1030	4.00	1.39–11.55	0.010 *
Postoperative LVEF	0.90	0.86–0.95	<0.001 *

Abbreviations: AISI—aggregate index of systemic inflammation, CI—confidence interval, LVEF—left ventricle ejection fraction, PAD—peripheral artery disease, SII—systemic immune-inflammatory index, SIRI—systemic inflammatory response index; *—statistically significant.

**Table 5 diagnostics-12-00634-t005:** Cox regression multivariable analysis.

Parameter	HR	95% CI	*p*-Value
Clinical characteristics:			
1. Stroke	3.39	1.06–10.91	0.040 *
2. PAD	3.83	1.27–11.56	0.017 *
Laboratory results:			
SII post > 952	3.44	1.02–11.66	0.047 *
Echocardiographic:			
LVEF < 45% postoperatively	4.11	1.21–13.95	0.023 *

Abbreviations: CI—confidence interval, LVEF- left ventricle ejection fraction, PAD—peripheral artery disease, SII—systemic immune-inflammatory index; *—statistically significant in the multivariable model, all HR are adjusted to other factors in the model.

## Data Availability

All data will be available under correspondence through e-mail address for three years following the publication after justifiable request.

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
