# Peer review of "The Significance of Systemic Immune-Inflammatory Index for Mortality Prediction in Diabetic Patients Treated with Off-Pump Coronary Artery Bypass Surgery"

_diagnostics, 2022, doi:10.3390/diagnostics12030634_

Round 1
Reviewer 1 Report
- The authors fail to address some limitations of their study. These should be clearly stated within the manuscript.
- Mortality as the main endpoint should be described in more detail. What type of mortality is this? It is not well-defined? How did you adjudicate mortality? This all should be stated and is lacking in the Methods section.
- No data on coronary artery disease burden are provided (e.g. number of diseased vessels, SYNTAX score, etc.)? These are extremely relevant?
- How does this score compare to EUROSCORE? Do you calculate EUROSCORE routinely for your CABG patients? I think this would add additional perspective to this work.
- Similarly, no data are provided on post-discharge medication which is a known factor contributing to post-discharge mortality outcomes.
- The introduction should be shortened to 1 1/3 page double-spaced at maximum.
- The regression model is poorly defined in the Methods.
Author Response
Dear Reviewer 1,
I would like to thank you on behalf of all co-authors for your valuable suggestions.
I’m grateful you found the time and engagement to help us improve our manuscript.
We followed your suggestions as presented below.
Kind regards
Tomasz Urbanowicz et. al.
Reviewer 1
- The authors fail to address some limitations of their study. These should be clearly stated within the manuscript.
Thank you for your valuable suggestion, we followed your advice.
We added more detailed limitation section.
- Mortality as the main endpoint should be described in more detail. What type of mortality is this? It is not well-defined? How did you adjudicate mortality? This all should be stated and is lacking in the Methods section.
Thank you for your valuable suggestion, we followed your advice:
The explanation on the mortality as a study endpoint was added.
The end-point analysis was based on all-cause mortality after Off-pump procedure. The analysis included any potential cause of primary point and was confirmed in national database.
- No data on coronary artery disease burden are provided (e.g. number of diseased vessels, SYNTAX score, etc.)? These are extremely relevant?
Thank you for your valuable suggestion, we followed your advice, the information has been already included:
Number of diseased vessels is presented in Results section.
Number of grafts is included in the Table 2.
- How does this score compare to EUROSCORE? Do you calculate EUROSCORE routinely for your CABG patients? I think this would add additional perspective to this work.
Thank you for your valuable suggestion, we followed your advice:
The Euroscore is calculated routinely and was not statistically different between the analyzed groups.
We added the information:
The Euroscore was calculated exclusively in each patient and presented no significant differences between groups 1.5 +/- 0.7 vs 1.6 +/- 0.8 in non-DM and DM groups (p=0.672), respectively.
Nevertheless, we are grateful for your comment, which influenced us to plan a concurrent study comparing different predictive scores to each other.
- Similarly, no data are provided on post-discharge medication which is a known factor contributing to post-discharge mortality outcomes.
Thank you for your valuable suggestion, we followed your advice:
Was added:
On discharge patients were treated with statin, antiplatelets, B-blockers and angiotensin-converting enzyme inhibitors.
As we are not able to provide a detailed information on long-term pharmacological treatment od individual patients and its’ influence on long-term mortality rate, we added that issue to the limitation section.
- The introduction should be shortened to 1 1/3 page double-spaced at maximum.
After your valuable suggestion regarding shortening of introduction, the following sentences were excluded
This glucose disorder is a significant risk factor for atherosclerosis development, including coronary artery disease []. Poor control of preoperative hyperglycemia in patients undergoing coronary artery bypass grafting may exert an adverse effect on long-term outcomes [].
Diabetes mellitus is characterized by chronic hyperglycemic episodes that significantly impact physiologic coronary vasodilatation []. It contributes to microvascular endothelium-dependent coronary impairment through oxidative stress and activation of inflammatory cascades [].
- The regression model is poorly defined in the Methods.
Thank you for your valuable suggestion, we followed your advice:
It was corrected by our statistic’s experts.
Kind regards
Tomasz Urbanowicz
on behalf of all co-authors
Reviewer 2 Report
Reviewer comments and suggestions
The study investigated the potential influence of postoperative inflammatory activation on mortality risk after off-pump coronary artery bypass grafting in diabetic patients. A total number of patients were 510 treated with off-pump coronary artery bypass grafting comprising 175 patients with type-2 DM The follow-up time was 3.7 +/- 1.5 years with a 9% all-cause mortality rate in the diabetic group.
The receiver operator characteristics curve analysis for postoperative systemic immune-inflammatory index (SII) showed significant (AUC=0.698, p=0.008), yielding a sensitivity of 68.75% and specificity of 71.07%. Hence, the authors suggested the role of perioperative inflammatory activation may play a crucial role in mortality prediction.
The paper has nicely written. However, the representation of the data and its description was poor not standard to reach the diagnostic journal. I would like to provide one more opportunity to go through all the below comments and modify your manuscript.
- Line 29 LVEF first time used in the abstract, required full form. I saw there were many inconsistencies related to abbreviations and the whole term described in the manuscript. Please check it again such as SII used in the paper.
- Line 48 modify TNF alfa with alpha
- Line 61-62 need to discuss more on the topic
- an “off-pump technique” need to discuss in the field of diabetes research
- Line 79 two typo errors, please modify
- Table 1 (No) representation for the number was not good.
- Line 94-96, it would be nice if the authors represent their study design in the form of a ray diagram
- I saw unnecessary lines were there in the manuscript like in sections 3.1, 3.2
- Line 153 the roc curve results were not perfect, it should be more than 70% for better outcomes.
- Line 213 table should be discussed concerning diabetic participants as per my understanding and more points needed to be explained here
- Table 4 needed to mention the adjustment here in legend
- Line 233-234 Why it is so?
- Line 244-246 The line should be mentioned whether the authors used in the study and mention references needed to be explained
- Line 281-285 The studies need to be discussed properly by relating their findings
- Comments for conclusion “long term prognosis of cvd risk (which was mention in multivariate analysis) better to consult the statistician or mortality or diabetes
- Please check the author's guidelines of MDPI, all journal formats needed to revise.
Author Response
Dear Reviewer 2,
I would like to thank you on behalf of all co-authors for your valuable suggestions.
I’m grateful you found the time and engagement to help us improve our manuscript.
We followed your suggestions as presented below.
Kind regards
Tomasz Urbanowicz et. al.
- Line 29 LVEF first time used in the abstract, required full form. I saw there were many inconsistencies related to abbreviations and the whole term described in the manuscript. Please check it again such as SII used in the paper.
Thank you for your valuable suggestion, we followed your advice:
In multivariable analysis, preoperative co-morbidities (stroke, peripheral artery disease, postoperative systemic inflammatory index >952, and postoperative left ventricle ejection fraction (LVEF) <45% were revealed as prognostic factors
It was corrected throught the manuscript: systemic immune-inflammatory index (SII).
- Line 48 modify TNF alfa with alpha
It was corrected.
- Line 61-62 need to discuss more on the topic
After your valuable suggestion regarding shortening of introduction, the following sentences were erased:
This glucose disorder is a significant risk factor for atherosclerosis development, including coronary artery disease []. Poor control of preoperative hyperglycemia in patients undergoing coronary artery bypass grafting may exert an adverse effect on long-term outcomes [].
Diabetes mellitus is characterized by chronic hyperglycemic episodes that significantly impact physiologic coronary vasodilatation []. It contributes to microvascular endothelium-dependent coronary impairment through oxidative stress and activation of inflammatory cascades [].
- an “off-pump technique” need to discuss in the field of diabetes research
The opinions on the superiority of off-pump or on-pump technique are conflicting. Some authors showed beneficial value of OPCAB in diabetic patietns
- Emmert MY, Salzberg SP, Seifert B, Rodriguez H, Plass A, Hoerstrup SP, Grünenfelder J, Falk V. Is off-pump superior to conventional coronary artery bypass grafting in diabetic patients with multivessel disease? Eur J Cardiothorac Surg. 2011 Jul;40(1):233-9. doi: 10.1016/j.ejcts.2010.11.003. Epub 2010 Dec 16. PMID: 21167727.]
while the other underlined superiority of on-pump mode
- Singh A, Schaff HV, Mori Brooks M, Hlatky MA, Wisniewski SR, Frye RL, Sako EY; BARI 2D Study Group. On-pump versus off-pump coronary artery bypass graft surgery among patients with type 2 diabetes in the Bypass Angioplasty Revascularization Investigation 2 Diabetes trial. Eur J Cardiothorac Surg. 2016 Feb;49(2):406-16. doi: 10.1093/ejcts/ezv170. Epub 2015 May 11. PMID: 25968885; PMCID: PMC4711702.
- Shroyer ALW, Quin JA, Wagner TH, Carr BM, Collins JF, Almassi GH, Bishawi M, Grover FL, Hattler B. Off-Pump Versus On-Pump Impact: Diabetic Patient 5-Year Coronary Artery Bypass Clinical Outcomes. Ann Thorac Surg. 2019 Jan;107(1):92-98. doi: 10.1016/j.athoracsur.2018.07.076. Epub 2018 Sep 28. PMID: 30273568.
We added the relevant information in the discussion section
- Line 79 two typo errors, please modify
We corrected the typos.
- Table 1 (No) representation for the number was not good.
We corrected the mistake
- Line 94-96, it would be nice if the authors represent their study design in the form of a ray diagram
We added the Flowchart of the study inclusion and explanation of exclusion criteria.
- I saw unnecessary lines were there in the manuscript like in sections 3.1, 3.2
Was corrected the lines.
- Line 153 the roc curve results were not perfect, it should be more than 70% for better outcomes.
Thank you for your valuable suggestion, we added preoperative and postoperative SII ROC analysis to enhance to better predictive value of postoperative results.
- Line 213 table should be discussed concerning diabetic participants as per my understanding and more points needed to be explained here
Thank you for your valuable suggestion, we followed your advice and added:
The multivariable model included parameters revealed as significant in DM group in univariable analysis
- Table 4 needed to mention the adjustment here in legend
It was corrected, by adding:
* - statistically significant in the multivariable model, all HR are adjusted to other factors in the model.
- Line 233-234 Why it is so?
Dear Reviewer, the multivariable analysis was based on preoperative history of stroke and diagnosis of peripheral artery disease regarded o lower extremities that’s why the conclusions were made as follow:
The multivariable analysis revealed a predictive effect of clinical preoperative factors referred to advanced atherosclerosis that resulted in a history of stroke episodes and peripheral artery disease.
- Line 244-246 The line should be mentioned whether the authors used in the study and mention references needed to be explained
Dear Reviewer, thank you for you valuable suggestion, the reference was added.
- Line 281-285 The studies need to be discussed properly by relating their findings
Dear Reviewer, we broadened our intentioned sentence .
The inflammatory cells and phenomena involved in atheroscleromatic disease burden include neutrophils, monocytes and macrophages activation. Currently, a large interest is laid in role of endothelial cell-derived extracellular vesicles in the progression of atherosclerosis. Undoubtfully, the inflammatory status, as a initiator and active participant of atherosclerosis development, should be taken into consideration as an important issue for monitoring and a potential therapeutic target.
- Comments for conclusion “long term prognosis of cvd risk (which was mention in multivariate analysis) better to consult the statistician or mortality or diabetes
Postoperative systemic immune-inflammatory index (SII) can be regarded as a significant but still weak predictive marker for long-term prognosis in diabetic patients treated by off-pump coronary artery bypass grafting.
- Please check the author's guidelines of MDPI, all journal formats needed to revise.
Kind regards
Tomasz Urbanowicz
On behald of all co-authors
Reviewer 3 Report
Thank you for the opportunity to review your manuscript entitled ,,The significance of systemic inflammatory index for mortality prediction in diabetic patients treated with off-pump coronary artery bypass surgery. ".
The parameter C-reactive protein was found to be an independent predictor of death and serious postoperative complications. CRP is a cyclic pentamer, belonging to the family of pentaxins – ligand-binding proteins in calcium-dependent reactions. Synthesis of CRP takes place primarily in the liver in response to pro-inflammatory factors Interleukin-1 (IL-1), Interleukin-6 (IL-6). CRP is one of the most important acute phase protein whose role is to participate in the body's immune response by facilitating the binding of complement. In the available literature, among others, that elevated values of CRP and IL-6 are predictors of inferior physical fitness and cognitive functions. Numerous studies have also documented that chronic inflammation is associated with increased morbidity and mortality. It has been described so far that factors such as diabetes, sedentary lifestyle, smoking or obesity are associated with elevated CRP values. However, not all patients with elevated CRP values are characterized by increased morbidity. The available literature suggests that morbidity in patients with elevated inflammation parameters results from an imbalance between proinflammatory and anti-inflammatory factors (1).
Frailty syndrome is a serious health problem for an aging population. With age, the susceptibility of the human body to the destructive influence of intra- and extra-environmental factors increases, while the body’s adaptability decreases. For a better understanding of the individual diversity of the rate of aging, the concept of frailty is used. Frailty syndrome is a consequence of a decrease in the physiological reserves of many organs. The factor responsible for the development of the frailty syndrome may be chronic inflammation(2). The presented study showed that the frailty syndrome diagnosed using the FRAIL scale indicating a reduced human physiological reserve may be one of the factors playing a decisive role in the qualification of patients to cardiac surgery. Noteworthy is the significant correlation between the FRAIL score and the CRP inflammation parameter, which may confirm that one of the causes of frailty syndrome is a chronic inflammatory process.
Were the severity of the frailty syndrome and CRP level assessed in the presented group?
Abstract, title and references.
The aim of the study is clear. The title is informative and relevant.
The references are relevant, recent, and referenced correctly.
Introduction.
It is clear what is already known about this topic.
The research question is clearly outlined.
Methods.
The process of subject selection is clear.
The variables are defined and measured appropriately.
The study methods are valid and reliable.
There is enough detail in order to replicate the study.
Results and Discussion.
The results are discussed from multiple angles and placed into context without being overinterpreted. The conclusions answer the aims of the study. The conclusions supported by references and results. The manuscript is well written and a stimulus for the readership.
Minor revisions:
Were the severity of the frailty syndrome and CRP level assessed in the presented group?
- Doi: 10.2217/bmm-2018-0101
- Doi: 10.2147/CIA.S239054
Author Response
Dear Reviewer 3,
I would like to thank you on behalf of all co-authors for your valuable suggestions.
I’m grateful you found the time and engagement to help us improve our manuscript.
We followed your suggestions as presented below.
Kind regards
Tomasz Urbanowicz et. al.
Reviewer 3
The parameter C-reactive protein was found to be an independent predictor of death and serious postoperative complications. CRP is a cyclic pentamer, belonging to the family of pentaxins – ligand-binding proteins in calcium-dependent reactions. Synthesis of CRP takes place primarily in the liver in response to pro-inflammatory factors Interleukin-1 (IL-1), Interleukin-6 (IL-6). CRP is one of the most important acute phase protein whose role is to participate in the body's immune response by facilitating the binding of complement. In the available literature, among others, that elevated values of CRP and IL-6 are predictors of inferior physical fitness and cognitive functions. Numerous studies have also documented that chronic inflammation is associated with increased morbidity and mortality. It has been described so far that factors such as diabetes, sedentary lifestyle, smoking or obesity are associated with elevated CRP values. However, not all patients with elevated CRP values are characterized by increased morbidity. The available literature suggests that morbidity in patients with elevated inflammation parameters results from an imbalance between proinflammatory and anti-inflammatory factors (1).
Frailty syndrome is a serious health problem for an aging population. With age, the susceptibility of the human body to the destructive influence of intra- and extra-environmental factors increases, while the body’s adaptability decreases. For a better understanding of the individual diversity of the rate of aging, the concept of frailty is used. Frailty syndrome is a consequence of a decrease in the physiological reserves of many organs. The factor responsible for the development of the frailty syndrome may be chronic inflammation(2). The presented study showed that the frailty syndrome diagnosed using the FRAIL scale indicating a reduced human physiological reserve may be one of the factors playing a decisive role in the qualification of patients to cardiac surgery. Noteworthy is the significant correlation between the FRAIL score and the CRP inflammation parameter, which may confirm that one of the causes of frailty syndrome is a chronic inflammatory process.
Were the severity of the frailty syndrome and CRP level assessed in the presented group?
Abstract, title and references.
The aim of the study is clear. The title is informative and relevant.
The references are relevant, recent, and referenced correctly.
Introduction.
It is clear what is already known about this topic.
The research question is clearly outlined.
Methods.
The process of subject selection is clear.
The variables are defined and measured appropriately.
The study methods are valid and reliable.
There is enough detail in order to replicate the study.
Results and Discussion.
The results are discussed from multiple angles and placed into context without being overinterpreted. The conclusions answer the aims of the study. The conclusions supported by references and results. The manuscript is well written and a stimulus for the readership.
Minor revisions:
Were the severity of the frailty syndrome and CRP level assessed in the presented group?
Dear Reviewer, thank you for your valuable suggestions. The analysis is retrospective so we cannot perform frailty evaluations.
The CRP was not routinely measured as this examination is performed in our department only when infection is suspected. However, the patients presenting with infection syndromes were excluded from the study.
According to your valuable comment we decided to launch prospective study that will include frailty syndrome
We based on simple parameters and indexes obtained from whole blood count parameters as possible predictors for all-cause mortality in our analysis.
Round 2
Reviewer 1 Report
Thank you for attending all of my comments.